# Urban Change in the United States, 1990–2010: A Spatial Assessment of Administrative Reclassification

**Bryan Jones** [1,2,*], **Deborah Balk** [1,2] **and Stefan Leyk** [3]

[1] CUNY Institute for Demographic Research, New York, NY 10010, USA; deborah.balk@baruch.cuny.edu

[2] Marxe School of Public and International Affairs, Baruch College, City University of New York, New York, NY 10010, USA

[3] Department of Geography, University of Colorado, Boulder, CO 80309, USA; stefan.leyk@colorado.edu

\* Correspondence: bryan.jones@baruch.cuny.edu

**Abstract:** In today's increasingly urban world, understanding the components of urban population growth is essential. While the demographic components of natural increase and migration have received the overwhelming share of attention to date, this paper addresses the effects of administrative reclassification on urban population growth as derived from census data, which remain largely unstudied. We adopt a spatial approach, using the finest resolution US census data available for three decennial census periods, to estimate the magnitude of reclassification and examine the spatial-temporal variation in reclassification effects. We supplement the census data by using satellite-derived settlement data to further explain reclassification outcomes. We find that while 10% and 7% of the population live in areas that underwent urban/rural reclassification during the 1990–2000 and 2000–2010 time periods, respectively (with smaller fractions of corresponding land), reclassification has a substantial effect on metrics derived to characterize the urbanization process—comprising roughly 44% and 34% of total urban population growth over each period. The estimated magnitude of this effect is sensitive to assumptions regarding the timing of reclassification. The approach also reveals where, how, to what degree, and, in some part, why reclassification is affecting to the process of urbanization on the fine spatial scale, including the impact of underlying demographic processes. This research provides new directions to more effectively study coupled nature–human systems and their interactions.

**Keywords:** urbanization; reclassification; urban population; urban growth

## 1. Introduction

As the world population becomes increasingly urban [1], it is important to understand how and when a population is identified as urban to fully understand urban growth and the implications of changes in an urban population. Such basic knowledge is presumed in each and every working definition of the urbanization rate. Yet, the processes that underlie the continuum of rural-to-urban transitions (and vice versa, where they occur) are difficult to expose, are based on different definitions across countries and time [1,2], and have not been the subject of significant inquiry [3]. In demographic terms, urban *population* growth—herein, urbanization—occurs as a function of (1) natural increase (i.e., births minus deaths) within urban and rural areas, and (2) migration between urban and rural areas. These two components have received the overwhelming share of attention in the understanding of urban change. However, administrative reclassification (The term administration reclassification is often applied because the decision to reclassify a census unit is an official administrative action taken by the appropriate national agency. It does not reveal anything regarding the nature of census units themselves, which are often simply enumerative as opposed to reflecting governance or administrative

function); the process of changing the official status of a given census unit or part of it (land, and the population living on it) from rural to urban (or more rarely, vice-versa) represents an important, yet largely understudied factor in this field of inquiry [4–7]. Consequently, little is known on how it affects estimates of urban growth that are commonly derived from census data. Moreover, because reclassification is an inherently spatial process [3], there is a knowledge gap regarding the impact of reclassification on the spatial growth of urban areas that may limit the ability to appropriately model and/or project the evolution of urban populations.

Studies of trends in, and the determinants of, urbanization typically either ignore the issue of reclassification, or combine the effects of migration and reclassification into a single term [8–10]. This is often justified by considering reclassification a nuisance to the demographic processes contributing to urban growth. A frequently cited figure attributing roughly 60 percent of urban growth to natural increase and 40 percent to migration and reclassification combined [11] is based on a comparative study of fewer than 40 countries across three decades, undertaken more than 20 years ago based on decadal change estimates up to 1980 [3,12], and thus should be treated with caution. Even more importantly, none of the existing investigations have used spatial methods to explore these issues, despite the fact that spatial processes are a critical component of changes related to reclassification. As such, the effects of reclassification may be significantly underestimated. In a recent study, Farrell's typology of urban growth [3], reproduced below in Figure 1, highlights the importance of a spatial dimension. Some reclassification occurs on the periphery of existing cities [7], or by the annexation of nearby localities into a larger metropolitan area, whereas some reclassification occurs as in-situ migration occurs [13]. The reclassification process is, in many cases, likely to be driven by the same demographic factors (fertility net of mortality and migration) that also operate in the areas that are not administratively reclassified, just disproportionately so, and may be accelerated by the redrawing of administrative boundaries. Yet, without a critical evaluation of such localities, it is impossible to know more about the magnitudes of these processes. For example, migration and reclassification may well have different causes, and involve populations of different age and sex compositions, and thus their relative importance to urban growth may change across the various stages of demographic and urban transitions [8]. It is possible, if not likely, that in some places, reclassification substantially contributes to what we measure as growth in the urban population, and the expansion of urban boundaries. And yet, models of urban change generally do not account for reclassification, and consider growth/expansion entirely as a function of migration and natural increase. This is a particularly salient problem to overcome in order to understand where, and why, urban growth occurs, the uncertainty in such estimates, and, subsequently, what is required to improve models of urban change and population projections that represent critical input to, for example, models predicting climate futures [14–16].

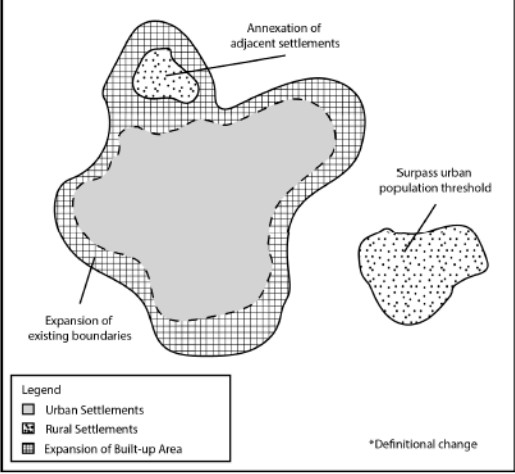

**Figure 1.** Farrell's [3] reclassification typologies based on the United Nations [17,18] and National Research Council [19].

The primary purpose of the paper is to use spatial methods to investigate the effect of urban/rural reclassification on the measurement of urbanization. The rationale for the method is that it reveals new knowledge that would aid the refinement of models for projecting urban change (population and land), critical components in the analysis of global change, that is potentially applicable in data-poor countries. Given that the majority of the future of population growth is expected to occur in data-poor regions, such as the cities and towns of the global South, it is important to identify and examine methods that enable the analyst to disentangle the components of observed magnitudes of urban growth from the effects of reclassification identified and quantified. Using the United States as a case study, here, we introduce a spatial method for assessing the potential role of reclassification in the observed level of urban change that may be implemented in other countries and for different time periods. Importantly, we are not suggesting that reported urbanization rates are incorrect but instead that there is likely some bias in the measurement of the relative contributions of migration and natural increase to the process of urbanization. Our case study provides an insight into the dimension of this hidden data-driven process using a combination of census data from three decadal censuses (1990–2010) and a remotely-sensed data product, the Global Human Settlement Layer (GHSL) representing built-up land at different points in time [20,21] to develop a better understanding of the urban process in general.

The enumerator area units of the US Census, blocks, are classified as either urban or rural by a set of criteria based on population size, density, and proximity to urban centers [22]. These changes in classification become effective at the completion of the decennial census, yet processes leading to this change could occur any time between two censuses. The temporal uncertainty associated with the reclassification has remained largely unaddressed but may well be very important for understanding aspects of demographic change (e.g., related to fertility, mortality, or migration). Further complicating matters is the fact that the census has changed its criteria for urban classification in each of the last three censuses [22]. This too presents some inherent complexities in our analysis, as discussed below. Yet, such changes in classification systems are the norm and are rather well documented. Moreover, the spatial distributions of growing urban populations resulting from either migration or reclassification can have distinctive patterns. Many studies of rural and urban transitions in the US use country-level data, which tend to remain stable over time, to avoid the complexities associated with reclassification. Nevertheless, they find increases in urban population in areas more distant from existing metropolitan centers (the "urban fringe") and in smaller cities lower in the urban hierarchy to be mainly driven by reclassification of previously rural areas [23]. In contrast, migration-driven urban growth occurs closer to central metropolitan areas where peri-urban areas transition to suburban land thus appearing as expansions of urban portions. In these locations, differential age patterns of migration have been observed [24]. For example, urban cores have been the destination of young migrants, but not migrants of other ages, and suburban areas were home to migrants of all ages. Recent research also shows heterogeneity across space and time that may suggest divergence from historical periods [25]. Understanding the spatial implications of how these processes drive reclassification will help the research community to better reflect and understand inherent sources of uncertainty in urbanization analysis.

The remainder of the paper is organized as follows. In Section 2 we introduce the relevant data, our methodology, and we discuss some of the challenges associated with characterizing reclassification. Section 3 reveals our results, beginning with those related to the census data and working from the national-level down to city-specific outcomes, highlighting Atlanta and New York City. Later in the section, we introduce GHSL-based findings, revealing the relationship between reclassification and the built environment. This latter piece is an important component of the potential applicability of these methods to other countries. In Section 4, we discuss the implications of our results, before drawing key conclusions in Section 5.

## 2. Materials and Methods

In this work, we take advantage of the high spatial resolution of block-level US census data from the 1990, 2000, and 2010 decennial censuses to investigate transitions occurring over two 10-year periods (1990–2000 and 2000–2010). For each period, the Census Bureau classifies blocks, and the population therein, as urban or rural according to characteristics of both the population and landscape. By overlaying block-level data from two successive census periods, we are able to quantify the degree to which population transitions between urban-rural classes. Additionally, because the data are organized over detailed spatial units, we are able to assess the spatial distribution of urban and rural people and pinpoint those areas in which transitions are occurring. Finally, we supplement the census data and definitions with a Landsat-based urban land product, the Global Human Settlement Layer (GHSL; [20,26]). A gridded layer at a 250-m spatial resolution, GHSL reports the portion of a grid cell that is "built-up" or comprised of man-made structures. We use these data to investigate the characteristics of census blocks, with particular interest in variation within and across urban/rural blocks as well as characteristics that may differentiate those blocks that experience a transition in census definition.

### 2.1. Census Data

The primary unit of analysis in this work is the census block, for which there is complete geographic coverage and a 100 percent population count (Summary File (SF) 1 data) over the last three decennial censuses (1990, 2000, and 2010) (Only SF1 data are available at block-level geographies. These data are publicly available at: https://www.census.gov/programs-surveys/decennial-census/data/datasets.2000.html). Census blocks are the smallest spatial units over which basic demographic data are compiled (population counts, age-structure, race, and gender), and are purely a census construct; that is, they are not representative of administrative units or governance. Blocks are delineated by both man-made and physical characteristics of the landscape, such as roads and rivers, and they vary in size across the urban–rural continuum, typically larger in rural areas and smaller in densely populated urban areas where they often comprise actual city blocks. Unlike larger census units, such as tracts, blocks vary widely in total population, ranging from zero to over a thousand in some densely populated city blocks. As spatial units, blocks themselves have been fairly unstable over time due to frequent boundary changes by the Census Bureau in response to development and shifting population. Consequently, the total number of census blocks has risen significantly over the previous three census periods, increasing from roughly 7 million blocks in 1990 to over 11 million blocks in the 2010 census. Further complicating matters, boundary changes are rarely as simple as dividing an existing block into smaller units, instead it is far more common that boundaries are simply redrawn such that new blocks consist of parts of multiple blocks from a previous period.

Census blocks, and the population contained within, are defined as urban or rural at each decennial census according to multiple criteria that have changed over time. Between 1950 and 1990 the primary building blocks for constructing urban areas (and thus delineating the urban population) were census incorporated or designated places (CDP; [22]). CDPs with a population of greater than 50,000 (often referred to as cities) were defined as urbanized areas (UAs). Beginning in 1990, the first census period for which there was complete geographic coverage at the block level, all blocks that fell in UAs were classified as urban. Additionally, any census blocks adjacent to such territory with a population density over 1000/mi$^2$ were included in the larger UA. Areas outside of UA blocks were defined as urban if they were part of a CDP with a population greater than 2500. For the 2000 census, CDPs were dropped as a starting point for constructing UAs, and census blocks were introduced as the primary building blocks. UAs were delineated as contiguous sets of blocks demonstrating a population density >1000/mi$^2$ and a total population of >50,000. In addition to UAs constructed from blocks, in 2000, the census introduced the urban cluster (UC) also constructed from contiguous blocks and adhering to the same population density requirement (>1000/mi$^2$), but requiring a total population between 2500–49,999. Any blocks falling within UAs and UCs were defined as urban, as were any blocks in close proximity to an UA or

UC (within 2.5 miles) provided their population density exceeded 500/mi$^2$. Finally, for the 2010 census, the urban definition was expanded to include certain blocks demonstrating industrial and commercial use (non-residential blocks demonstrating substantial urban land-cover) located in close proximity (within 0.25 miles) to populated urban blocks in UAs and UCs. Urban land-cover (impervious surface) was delineated using the National Land Cover Database [20].

## 2.2. The Global Human Settlement Layer (GHSL)

The Global Human Settlement Layer, produced by the Joint Research Center (JRC) of the European Commission, is a remote sensing-derived global data product that represents built-up land for different points in time over 40 years (1975, 1990, 2000, and 2014) at a fine spatial resolution (approximately 38 m, aggregated to 250 m). The original resolution data are binary, indicating either the presence or absence of a built structure in each 38m grid cell [20,26,27]. The aggregated data are constructed from the 38m cells to quantify the percentage of each cell that is built-up. This construction implies an aggregation of the original data to 304m and a subsequent resampling step to 250m to facilitate compatibility with other 1 km global land cover and population data products. A recent validation study has generally confirmed the accuracy of the GHSL data layers for the different points in time in urbanized settings but also reported higher levels of classification errors in rural regions (for details, see [28,29]).

## 2.3. Methods

We begin with population counts and urban/rural status organized spatially at the block-level for the whole of the United States at each of the previous three decennial census years (1990, 2000, and 2010). Because boundary changes between censuses at the block-level are frequent (affecting some 60% of blocks between each time period), it is necessary to derive spatially consistent layers to facilitate our analysis of change over each of the two 10-year periods in this study (1990–2000, 2000–2010). To do so, we overlay the block boundary layers from the beginning and end point of each period (e.g., 1990 and 2000) and derive the spatial union, resulting in a new layer that respects the boundaries of both periods. This process has the effect of dividing many blocks into smaller sub-block units which we term "block bits". We then populate the blocks and block bits in this new spatial layer by allocating population counts and urban/rural status from both the beginning and end point of each period (e.g., 1990 and 2000) to each unit in the new layer. In cases where the union function created block bits, we allocate population proportionally according to land area (e.g., the area of the block bit as a portion of the block from which it was derived in both the beginning and end period) and urban/rural status according to the status of the parent block from each time period. We carry out this process twice (1990–2000 and 2000–2010), resulting in two new spatial layers from which we can analyze transitions over the first (1990–2000) and second (2000–2010) census periods.

For each period we classify blocks and block bits into one of four categories based on urban/rural status at the beginning and end of each period: (1) transition from urban-to-rural, (2) transition from rural-to-urban, (3) remained urban, and (4) remained rural. We then aggregate the population in each class from blocks/block bits up to city, state, census division/region, and national scales, and compare both population totals and population change over each census period. Additionally, we produce maps to assess spatial patterns in the distribution of classes as well as population change within and across classes, focusing in particular on major urban areas and their hinterlands (areas likely to exhibit evidence of transitions between rural and urban classes over time). To add to this analysis, we considered the degree to which blocks and block bits are "built-up" using the GHSL. For each spatial unit, we calculate the average GHSL value, and from there the range, mean, and median GHSL value for each class at different spatial scales, again paying attention to both values at the beginning and end of each period and change over time.

Modeling the Timing of Reclassification

Census-based urban classifications change over time for various reasons, as discussed above. Officially and practically, a change in urban status occurs and is only evident at the completion of a census survey, when census units are assessed against the urban standard. Realistically, however, for those blocks redefined as urban from rural (or, though less likely, in the opposite direction), the actual timing of the transformation is not known. It is very likely, however, that for each block—or block bit—in question, the standard for urban/rural reclassification is met at some point over the 10 years between censuses as opposed to the conclusion of the census period. How we choose to conceptualize the timing of reclassification will influence substantially the degree to which it impacts the measurement of overall urbanization.

Three stylized possibilities are shown in Figure 2 below using the 2000–2010 census period as an example. Recalling that the primary mechanism for classifying blocks as urban/rural is population density (while population agglomeration and proximity and land cover characteristics also contribute), consider three hypothetical blocks at census time (t) with a density below the required 1000/mi$^2$ to be considered urban (thus they begin the period as rural blocks). As time advances towards census period (t+1), the population density of Block *a* rises quickly, then slowly levels off. Block *b*, in contrast, grows slowly in the early part of the period before increasing in density rapidly just before period (t+1). Block *c* does not grow at all but exhibits a substantial amount of impervious surface and is located in close proximity to other densely populated blocks. From the census perspective, all three of these blocks are reclassified as urban at time (t+1); Blocks *a* and *b* because they meet the density threshold, and Block *c* because of a definitional change regarding impervious surface and proximity. However, from a more realistic perspective there was substantial temporal variation in the reclassification process. Block *a* met the urban criteria shortly after time (t), while Block *b* technically remained rural much longer, meeting the threshold near the end of the period. Block *c* did not meet the definition of urban until the end of the period, when a more inclusive urban definition was introduced. The question of timing is non-trivial in terms of the portion of urbanization measured that can be attributed to reclassification. If we assume that the transition occurs at the end of the period, then all of the population growth in Blocks *a* and *b* is considered to have occurred in a rural block, thus the entire population is counted as "reclassified".

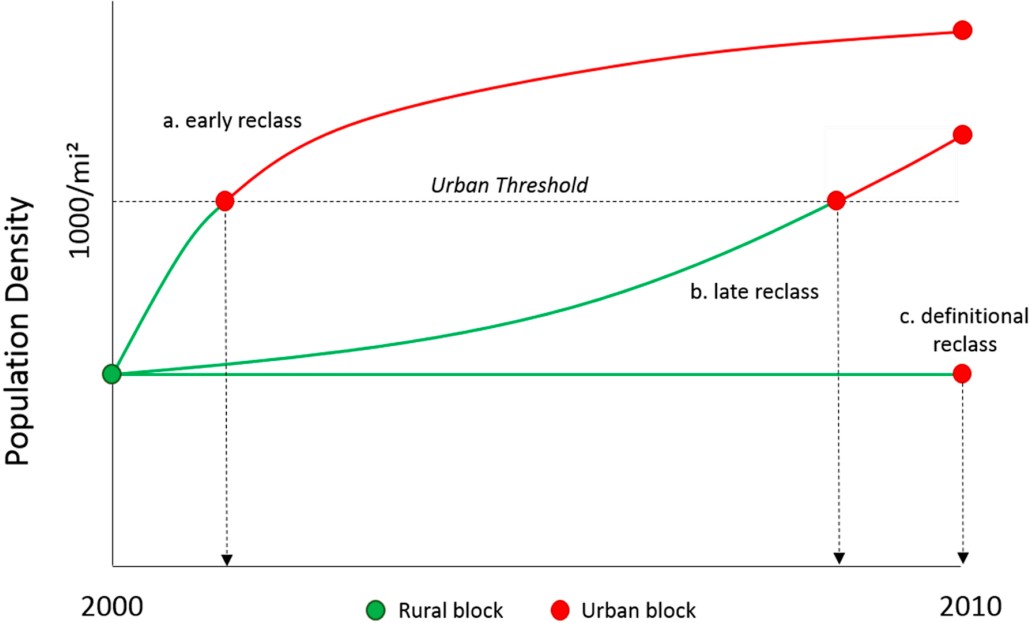

**Figure 2.** Stylized example of three alternative pathways to rural-to-urban reclassification over the 2000–2010 census period.

As this Figure demonstrates, the exact timing is unknown, causing a high degree of uncertainty in the attribution of reclassification versus population growth itself and the respective contribution to the measurement of urbanization. In this paper, we explicitly consider the impact of different assumptions regarding timing, report on the full range of potential outcomes, and offer suggestions for dealing with the data limitations at the block level during intercensal periods. From the stylized example, we can bookend the relative effect of reclassification on the assessment of total urbanization by considering the two most extreme cases. In the first, we assume that all reclassification that will take place occurs immediately following the beginning of the time period, in this case the 2000 census. In this case, all persons in a block that undergo a transition are reclassified in place, and all subsequent natural increase and net migration that occurs is accounted for as such. This method will produce the lowest estimates of the relative effect of reclassification. On the other extreme, we assume that all reclassification takes place at the end of the census period when (typically) new definitions are applied upon completion of the census count—here, occurring in 2010. From a practical standpoint, as noted above, this is how the census operates, and thus is the existing procedure, although there are no records of the census or any other interested party extracting reclassification estimates from these data. Logistically, adopting this approach means that the natural increase and migration that occurs during the time period in a block that is reclassified is counted as reclassification. This approach will produce the largest effects of reclassification to measuring urbanization and will often minimize the impact of natural increase and migration. Given that we do not know the exact timing of reclassification at the census block level, but do know that the processes illustrated in Figure 2 are more indicative of reality than either of the prior two cases, in a third approach we will assume that, for all blocks reclassified over a 10-year census period, the timing of actual reclassification (when the definitional thresholds are met) is spread uniformly across the period. It should be noted that those different scenarios described may be applied to whole census blocks but even more often to block bits further complicating the matter, analytically, but following the same logic.

## 3. Results

Our analysis is divided into two primary categories: (1) census-oriented and (2) satellite-oriented. The census-oriented analysis examines changes between official urban–rural designations. The satellite-oriented analysis examines changes in built-up area for all possible transitions between these census designations. Each subsection is further disaggregated across geographies, and in the case of the latter, the satellite data are combined with spatial census data to enhance the analysis.

*3.1. Census-Oriented Analysis: Transitions between Official Rural and Urban Designations*

In this section, we present results derived only from our investigation of census-data (both spatial and aspatial) over the period 1990–2000 and 2000–2010. We start at the national-level, work down to census regions/divisions, and finally look specifically at two cities: Atlanta and New York.

3.1.1. National-Level

We begin by looking at national-level results, including the population in each of the four categories at the beginning and end of each of the two census periods (1990–2000, 2000–2010), population change by each category, and estimates of the contribution of reclassification to measuring urbanization over each census period. Table 1 includes population and population change figures for both periods. Trends are similar across each period, even if the magnitude of change varies somewhat. Large portions of the population are in blocks that do not change classification, "remained urban" and "remained rural", which are comprised of some 183–231 million (urban) and 49–57 million (rural) persons. Among rural blocks, population growth varies substantially between those blocks that remain rural (14% and 7% between 1990–2000 and 2000–2010, respectively) and those that transition to urban (108% and 253% between 1990–2000 and 2000–2010, respectively). Over both periods, population growth in blocks transitioning from rural-to-urban was roughly 13 million, outstripping the growth in blocks

that remained rural substantially, despite comprising a much smaller geographic area. Only a small number of blocks transition in the opposite direction (urban-to-rural), and in both periods this class of blocks experiences aggregate population loss.

**Table 1.** Aggregate urban/rural population totals and change for four transition types, USA, 1990–2000 and 2000–2010.

| Transition Type | Population Estimates | | | |
|---|---|---|---|---|
| | Based on Status at Beginning of Decade | Based on Status at End of Decade | Change | % Change |
| **1990–2000** | | | | |
| Rural-to-Urban | 12,230,403 | 25,449,191 | 13,218,788 | 108.08% |
| Urban-to-Rural | 3,633,651 | 2,716,676 | –916,975 | –25.24% |
| Remained Urban | 183,417,909 | 196,910,455 | 13,492,546 | 7.36% |
| Remained Rural | 49,425,229 | 56,344,201 | 6,918,972 | 14.00% |
| Urban Total | 187,051,560 | 222,359,646 | 35,308,086 | 18.88% |
| Rural Total | 61,655,632 | 59,060,877 | –2,594,754 | –4.21% |
| **2000–2010** | | | | |
| Rural-to-Urban | 5,251,441 | 18,520,310 | 13,268,869 | 252.67% |
| Urban-to-Rural | 3,441,570 | 2,014,919 | –1,426,651 | –41.45% |
| Remained Urban | 218,913,517 | 230,731,464 | 11,817,947 | 5.40% |
| Remained Rural | 53,808,728 | 57,477,054 | 3,668,326 | 6.82% |
| Urban Total | 222,355,087 | 249,251,774 | 26,896,687 | 12.10% |
| Rural Total | 59,060,169 | 59,491,973 | 431,804 | 0.73% |

Note that the 2000 population in each class does not match between the first period (1990–2000) and the second period (2000–2010) because the sub-block spatial units over which the analysis was conducted vary as a function of the underlying block-level geometry changes (i.e., 1990–2000 is the union of 1990 and 2000 block geometries, while 2000–2010 is the union of 2000 and 2010 geometries). So, for example, the population in a block bit that transitions from rural-to-urban during the 1990–2000 and then remains urban during 2000–2010 would contribute to the 2000 Status "Urban" on the Rural-to-Urban row for 1990–2000, and the 2000 Status "Urban" on the Remained Urban row for 2000–2010.

To understand the implications of these results for the contribution of reclassification on measuring urbanization, we consider the portion of total urbanization over each period that is accounted for by population undergoing a change in urban/rural status. As noted, assumptions regarding the timing of these changes weigh heavily. For example, if in the 1990–2000 period we assume that reclassification takes place immediately following the 1990 census, then some 12,230,403 people would transition from rural-to-urban status, while 3,633,651 would experience the opposite transition. To calculate the effect of reclassification on measuring urbanization, we take the number of people reclassified as urban, less the number reclassified as rural, and divide by the total change in the urban population over the period (35,308,086). In this case, reclassification would comprise roughly 24% of total urbanization. On the other extreme, assuming reclassification occurs just prior to the 2000 census, some 25,449,191 persons transition from rural-to-urban, while only 2,716,676 are reclassified as rural. In this case, the reclassification effect rises to over 64%. The more realistic assumption that blocks reach the definitional threshold necessary for reclassification in a somewhat uniform distribution over the 10-year period thus leads us to believe that reclassification is responsible for roughly 44% of measured urban population growth between 1990 and 2000, with the remaining 56% coming from natural increase and net-migration (which includes natural increase and net-migration in blocks that remained urban and those that were reclassified). The range of outcomes for the period 2000–2010 is wider than the earlier period, stretching from only 7%, if we assume reclassification occurs at the beginning of the period, to 61% if we assume the opposite, with a contribution of roughly 34% to the measurement of total urbanization if we adopt the more realistic uniform transition assumption. In general, population change is far more pronounced in those areas reclassified as either urban or rural, while in blocks that are not reclassified, population grew modestly on aggregate.

A number of factors and processes lead to these results and untangling the interacting parts is necessary to understand exactly how and why reclassification occurs. Here, we focus primarily on the rural-to-urban component as this is an important piece of the urbanization process. At the beginning of each census period there are a set of urban and rural blocks. Typically, most of the rural blocks remain rural, but a small subset will transition to urban over each period. At the end of each period we therefore have a set of urban blocks or block bits that were (1) urban at the beginning of the census period and (2) transitioned to urban during the time period. The contribution of the reclassification to the measurement of urbanization is a function of two factors: the amount of population change occurring in the blocks and block bits that transitioned during the period relative to those that were already urban, *and* the assumed timing of reclassification to urban in those units. Regarding the latter, if we assume that reclassification occurs at the beginning of the period, the population growth in each block bit can be attributed mostly to migration or natural increase. In other words, the portion of urbanization resulting from net migration and natural increase will be larger relative to reclassification. If we assume that reclassification occurs at the end of the period, then any children that were born during this time period or migrants relocating to these areas are technically considered rural until before the end of the period. Then, they are reclassified as urban at the end of the census period, and, as such, the portion of urbanization ascribed to reclassification will be larger relative to other factors. In reality, each block bit that is reclassified technically meets the definitional threshold for becoming urban at some point during the 10-year period. As such, the assumption that the temporal distribution of blocks and block bits meeting the threshold is uniform over the 10-year period can be seen as the more objective one because it essentially assumes that the average block is reclassified at the midpoint of the census period.

### 3.1.2. Census Regions and Divisions

To look for regional variation, we further disaggregated population change by class over each census period at the census-region and -division levels. Population and population change by class for each geography can be found in Tables S1–S4 in the supplementary information, while the relative contribution of reclassification to measuring urbanization under different timing assumptions by census regions and divisions is included in Table 2. At alternative geographies, the range of outcomes resulting from the different assumptions on timing is always quite large, in some cases encompassing nearly the full change in the urban population (e.g., the Midwest Region and its divisional components between 2000 and 2010). In such cases, the bulk of the urban population growth is taking place in blocks and block bits that are reclassified from rural-to-urban (e.g., see the Midwest Region and East North Central division in Table 2). In the East North Central Division blocks that begin as and remain urban actually lose population on aggregate. Thus, regionally, the assumption on timing can be the primary determinant of the role of reclassification. Under the assumption of uniform transition, the reclassification effect ranges from 25% (Mountain division) to 79% (New England Division) in the earlier census period, and from 20% (Pacific division) to 67% (East North Central Division) in the later period. Despite substantial variation around the national average over each period, there is a fair degree of consistency within census regions and divisions when comparing 1990–2000 with 2000–2010. The exception is the Northeast Region and its divisional components (New England, Mid-Atlantic) in which the reclassification effect in the second period is less than half that of the first, owing to a much smaller number of blocks (and thus people) being reclassified in the latter period. In general, population change in blocks of the urban-to-rural class is consistently lowest in the Northeast and highest in the West (see Tables S1–S2). Growth in blocks remaining urban or rural is larger in the South and West, reflecting higher growth rates in those regions relative to the Northeast and Midwest. Finally, reclassification is a larger contributor to urban growth in the 1990–2000 period, regardless of assumptions about timing.

**Table 2.** Estimated contribution of reclassification to urbanization by assumption regarding the timing of reclassification, the national-level, and census regions/divisions during the 1990–2000 and 2000–2010 census periods.

| Decade: | 1990–2000 | | | 2000–2010 | | |
|---|---|---|---|---|---|---|
| Timing Assumption: | Begin | Middle | End | Begin | Middle | End |
| National | 24.3% | **44.4%** | 64.4% | 6.7% | **34.0%** | 61.4% |
| Northeast | 53.7% | 68.7% | 83.6% | 9.4% | 31.7% | 54.1% |
| New England | 65.0% | 79.1% | 93.1% | 11.0% | 33.1% | 55.2% |
| Mid-Atlantic | 49.5% | 64.8% | 80.1% | 8.9% | 31.3% | 53.7% |
| South | 22.7% | 44.8% | 66.8% | 9.0% | 34.8% | 60.5% |
| South Atlantic | 29.0% | 50.2% | 71.3% | 10.5% | 33.8% | 57.1% |
| East South Central | 19.5% | 52.2% | 84.9% | 14.0% | 48.4% | 82.9% |
| West South Central | 10.2% | 31.2% | 52.3% | 5.1% | 32.5% | 59.9% |
| Midwest | 31.1% | 55.8% | 80.6% | 10.0% | 55.2% | 100.5% |
| East North Central | 38.6% | 62.2% | 85.8% | 11.4% | 67.6% | 123.8% |
| West North Central | 7.4% | 35.8% | 64.2% | 8.4% | 41.7% | 75.1% |
| West | 8.8% | 26.1% | 43.4% | 1.4% | 26.8% | 52.1% |
| Mountain | 6.5% | 25.9% | 45.2% | 3.9% | 34.9% | 65.8% |
| Pacific | 10.3% | 26.2% | 42.2% | -0.5% | 20.5% | 41.5% |

### 3.1.3. City-Scale Census Analysis: Atlanta and New York

At the national-level recall that the majority of census units fall into one of the two classes that remained unchanged (urban or rural). We also find that, by total area, the majority of the country's land area falls into the "remained rural" class, and transitioning classes, and urban block bits are typically smaller and concentrated. By examining specific cities, we are able to better visualize and interpret fine-scale trends in classification over time. Here, we focus specifically on Atlanta and New York City, though bearing in mind that there is much heterogeneity in urban form, and change thereof, among the cities in the US [30]. The former is representative of cities of the sunbelt region that have experienced significant growth and expansion over the past half century. The latter is a very large urban area, in the less dynamic (in terms of population change) Northeast, that has been very densely populated for centuries and is significantly constrained geographically by the coastline.

Figure 3 illustrates the spatial distribution of transitional classes for Atlanta (Figure 3a,b) and New York City (Figure 3c,d) for the periods 1990–2000 and 2000–2010, respectively. Atlanta demonstrates a somewhat classic, monocentric [31] urban structure, a roughly circular urban core expanding in all directions. This characteristic is illustrated by the rural-to-urban block bits distributed around the urban fringe in both time periods, although with more intensity in the 1990–2000 period. There are noticeable rural-to-urban transitions occurring in the Northeast portion of Atlanta towards Gainesville, GA, between 1990–2000, followed by infill of the area between Gainesville and Atlanta after 2000. Also of note is the growing urban corridor between Atlanta and nearby Athens, GA, to the East.

By contrast, the New York metropolitan area is a long-established, large urban area that includes New York City proper, a large number of smaller cities that are fully connected to the primary urban agglomeration, and a collection of smaller satellite cities. Transitions from rural-to-urban are evident on a Northeast to Southwest axis, particularly in the first period, extending along the coast of the Long Island Sound into Connecticut and towards Philadelphia in the opposite direction. In the latter period, which does include the financial crises that hit the region fairly hard, much less rural-to-urban transition is noticeable. Also evident in the first period are a few relatively large areas of urban-to-rural transition. Contrary to what one might expect, these transitions are not primarily driven by population loss or urban decline, but instead by boundary changes in which larger blocks were subdivided into smaller units (block bits), often separating urban settlements from adjacent rural areas (including parks or forested land). Recall that the number of blocks at the national level has continued to expand over each successive census period. This trend is driven in part by census efforts to improve the distribution

of blocks such that each unit is more ubiquitous in nature (e.g., entirely urban or rural rather than a mix of the two types of development).

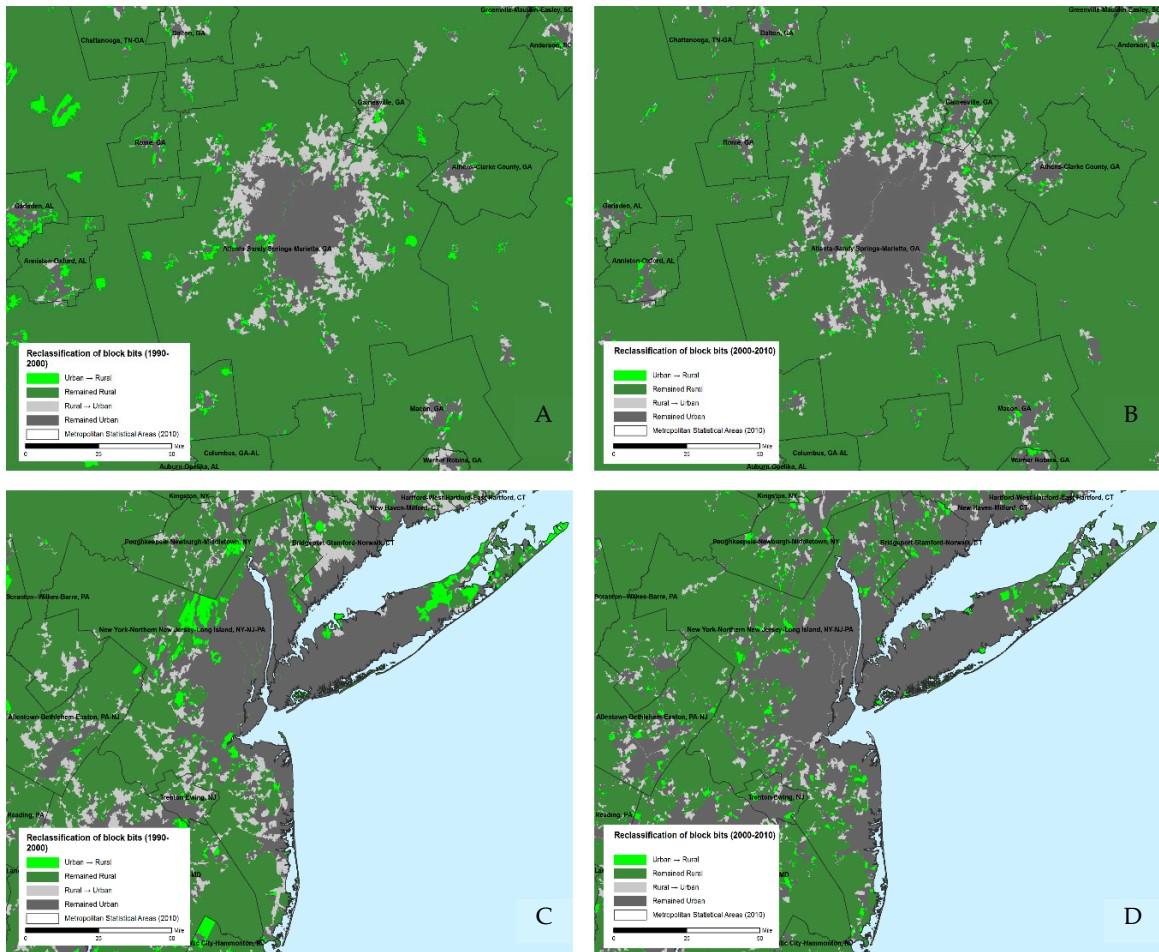

**Figure 3.** Distribution of land by transition type for Atlanta (**A**,**B**) and New York City (**C**,**D**) for 1990–2000 (**A**,**C**) and 2000–2010 (**B**,**D**).

Figure 4 illustrates population change by block bit and transition class for Atlanta over the period 1990–2000. Several noticeable patterns are evident. In Figure 4A, we see that the bulk of the growth in areas that remained rural took place on the urban fringe. Unsurprisingly, we also find that most areas transitioning from urban-to-rural (Figure 4C), which are generally located on the urban fringe, also grew substantially over the period. Among those areas that remained urban, we also find that population growth is skewed towards the outer boundaries of the city, suggesting that the urban fringe, including transitioning block bits and nearby urban/rural areas, comprise what is clearly the most dynamic area of the city. Figure 4D reveals that there is no clear pattern of population change in block bits transitioning from urban-to-rural (by far the smallest class by land area and population). In some cases, there is significant population loss, and in other cases, we find large gains. It is likely that these block bits are indicative of several phenomena, including actual population loss (areas experiencing urban decline), as well as artifacts of boundary changes that separate rural and urban areas of blocks that were single units in 1990, but have been redefined and split into multiple units in 2000. It can be particularly difficult to accurately calculate population change in such cases, because we do not know exactly how block-level population is distributed with blocks. This remains an area for future research.

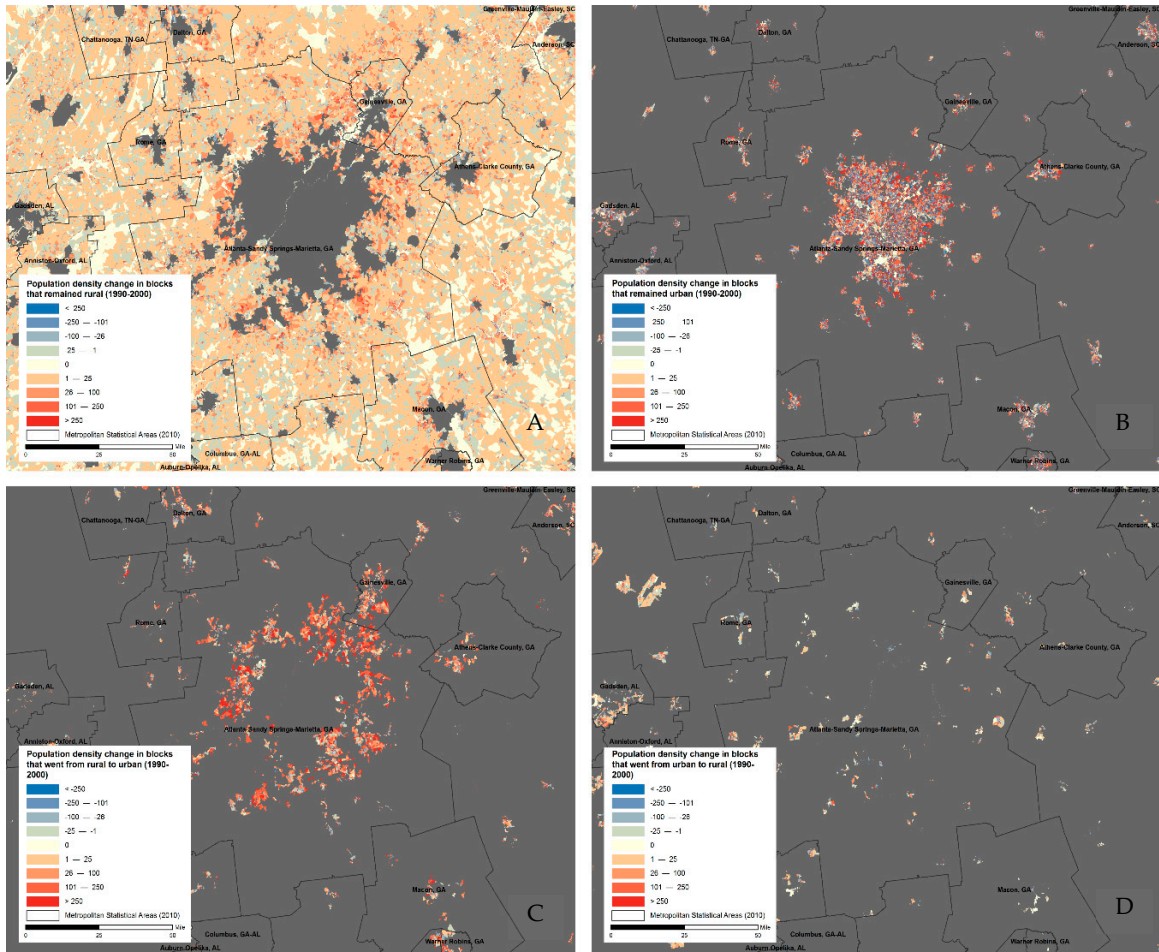

**Figure 4.** Population change (1990–2000) by block bits for (**A**) remained rural, (**B**) remained urban, (**C**) rural-to-urban, and (**D**) urban-to-rural transition classes.

### 3.2. Satellite-Oriented Analysis: Built-Up Area Changes within Census-Class Transitions

Rapid improvement in remotely sensed/satellite data processing and production provides increasing opportunities to supplement census-based spatial data to advance our understanding of urban processes. Thus far, we have examined aggregate, and spatially explicit patterns of urban/rural population classes and transitions, and assessed patterns of population change within and across each of our transition classes. We now supplement this analysis by analyzing changes in the built environment of these different classes, using one of the recently developed, global, built-up land products developed from Landsat imagery; the global human settlement layer (GHSL). Figure 4 illustrates the distribution of built-up land of varying intensity across the greater Atlanta area at four points in time: 1975 (Figure 5A), 1990 (Figure 5B), 2000 (Figure 5C), and 2014 (Figure 5D). The relatively uniform expansion of Atlanta since 1975 is clearly visible, as are intensifying urban transportation corridors between Atlanta and smaller satellite cities.

To assess the proportion of built-up land across the different transition classes for the continental United States in 1990, 2000, and 2010, we computed the average built-up area over each class, that is, the average portion of the land area that is built-up at the beginning and end of each of the two periods (substituting 2015 for 2010 in the latter period). In doing so, the GHSL data tell us more about how each class may vary from the others in terms of the landscape, as well as what types of changes in urban land use (or lack thereof) are occurring. Additionally, it may also be possible to identify characteristics of areas at one point in time that indicate a likely transition in the next. The national-level results are presented in Table 3 (results for census regions and divisions can be

found in Tables S5–S8). Unsurprisingly, the class that remained urban exhibits the highest built-up percentages, ranging from 42% to 48%. These areas also experienced the slowest growth, however, suggesting that built environment is well established and that the building density is saturated to some degree, a narrative that fits with the slower rates of population growth evident in Tables 1 and 2. In contrast, newly designated urban areas (rural-to-urban) grow at three to six times the rate of already urbanized areas (22%–41%), however, these areas, typically on the urban fringe, still exhibit roughly half the built-up proportion as the urban core. Perhaps the least intuitive result is related to areas reclassified from urban-to-rural. While substantially less built-up than land that remained urban or land that transitioned from rural-to-urban, areas reclassified as rural did exhibit growth, of roughly 25% in the built-up area, over both periods. There are two likely explanations for this pattern. First, as boundaries are redrawn to improve the census distribution of blocks, areas that are commercial or institutional in nature (such as shopping centers or industrial facilities) are often separated into their own blocks. If located away from the urban core, these areas will not meet the proximity or population thresholds necessary to be defined as urban, and thus will be classified as rural despite the built-up signal. Second, and also related to boundary changes, blocks that display both suburban and rural characteristics along the urban fringe are often split to reflect more ubiquitous characteristics. In such cases, new blocks that exhibit smaller populations may be redefined as rural but at the same time reflect the type of increasing development indicative of the urban/rural fringe. Finally, areas remaining rural also exhibit growth in built-up land over both periods, but the built-up level remains very low on average (less than 0.5%).

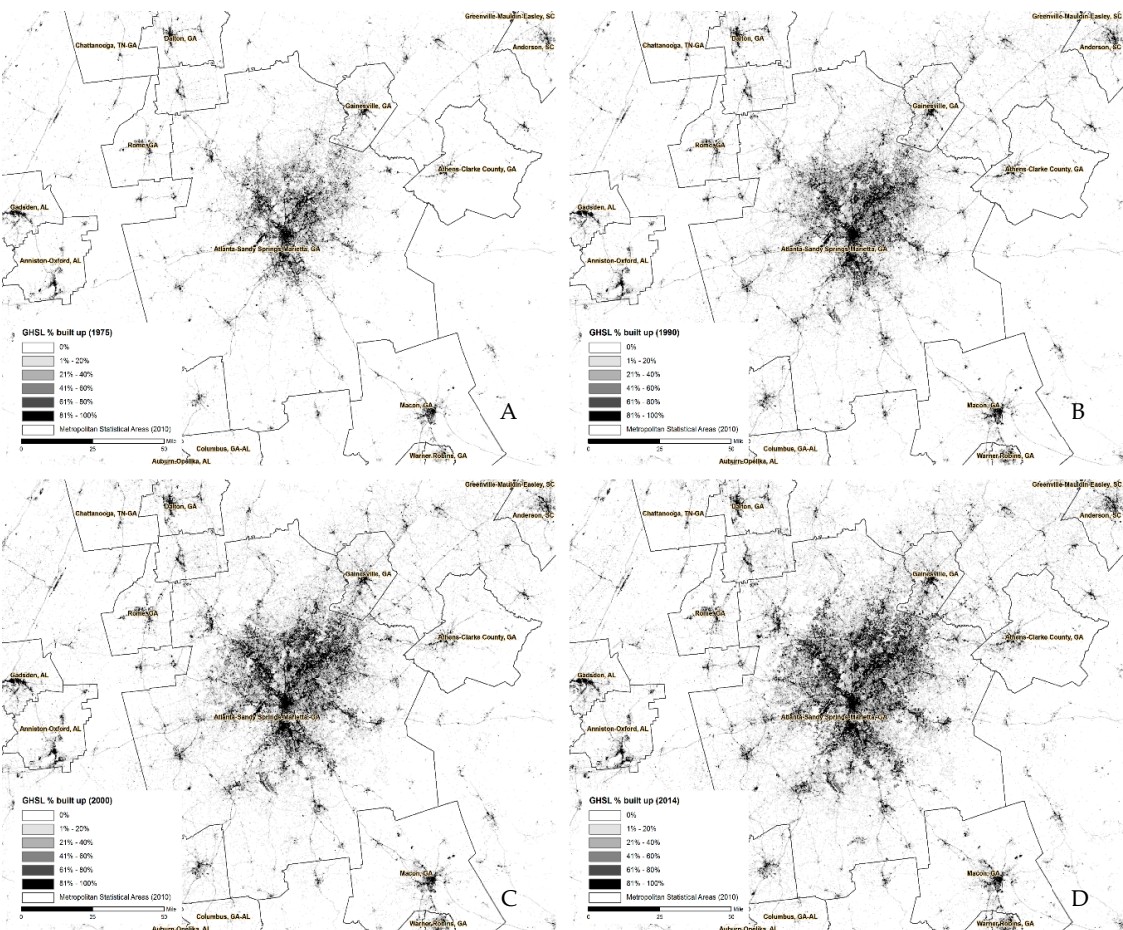

**Figure 5.** Intensity of built-up land area, as indicated by the Global Human Settlement Layer (GHSL) for the metro Atlanta area in (**A**) 1975, (**B**) 1990, (**C**) 2000, and (**D**) 2014.

**Table 3.** Proportion built-up area, as detected by the Global Human Settlement Layer (GHSL) by transition class, 1990–2000 and 2000–2010.

| | Proportion Built-Up | | | |
|---|---|---|---|---|
| **Transition Type** | **At Beginning of Decade** | **At End of Decade** | **Change** | **% Change** |
| **1990–2000** | | | | |
| Rural-to-Urban | 0.2006 | 0.2452 | 0.0446 | 22.2% |
| Urban-to-Rural | 0.0716 | 0.0903 | 0.0187 | 26.2% |
| Remained Urban | 0.4422 | 0.4774 | 0.0352 | 8.0% |
| Remained Rural | 0.0032 | 0.0042 | 0.0010 | 32.6% |
| **2000–2010** | | | | |
| Rural-to-Urban | 0.1618 | 0.2282 | 0.0664 | 41.0% |
| Urban-to-Rural | 0.0680 | 0.0841 | 0.0162 | 23.8% |
| Remained Urban | 0.4265 | 0.4598 | 0.0333 | 7.8% |
| Remained Rural | 0.0033 | 0.0045 | 0.0011 | 34.0% |

One of the primary goals of this work is to assess the usefulness of products such as GHSL as a refined proxy for urbanization/reclassification and, more broadly, to identify potential leading indicators that suggest that reclassification is likely in the near future, with a particular emphasis on the rural-to-urban transition. Figure 6 plots the average value of built-up proportions for each of the nine US Census Divisions for blocks that were rural in 2000 and either remained rural (green dots) or transitioned to urban (red dots) against observed population change over the period 2000–2010. Rural blocks that transitioned to urban exhibited much higher levels of built-up proportion (10%–23%) than those that remained rural (0%–1%). Blocks that transitioned also experienced a population growth of 50% to nearly 700% (on aggregate at the divisional level), while those that remain rural experience a population growth of less than 10%. These results suggest that GHSL is a potentially useful leading indicator of blocks that are likely to transition from rural-to-urban, even at relatively low built-up levels (when compared to existing urban blocks).

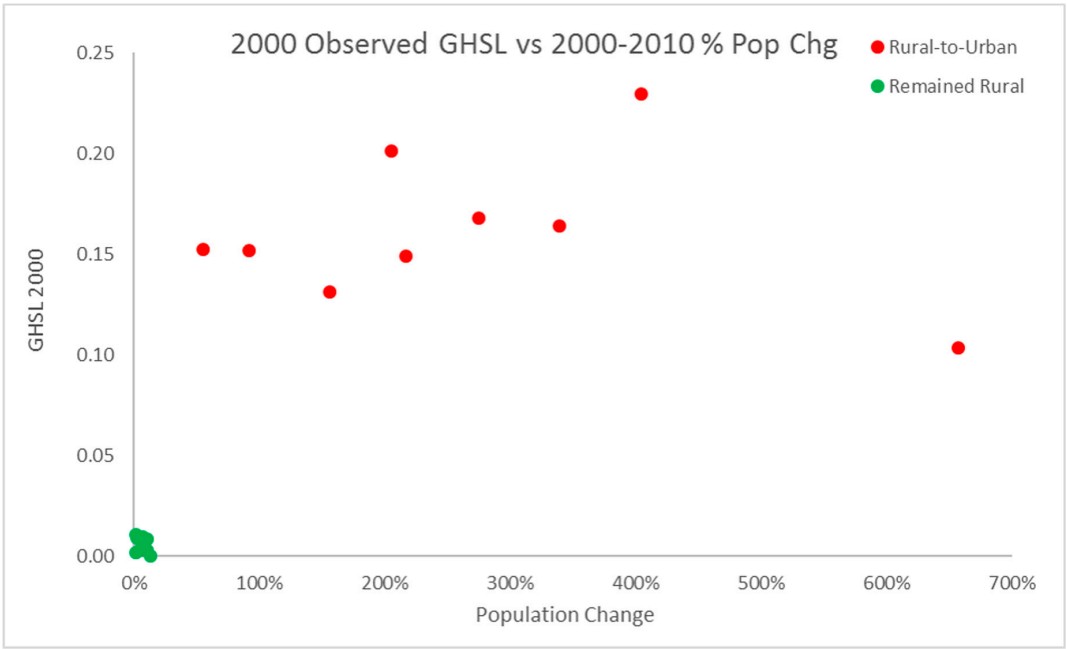

**Figure 6.** Built-up proportions for each of the nine US Census Divisions of transitions among rural blocks in 2000 versus % population change from 2000–2010, comparing those that transitioned to urban vs. those that remained rural.

## 4. Discussion

The methods for estimating and assessing the effect of reclassification on estimates of urban growth/change is particularly useful for several reasons. To gain a better understanding of the degree to which the different demographic forces are driving urbanization, one must understand the process of administrative reclassification. Recall that urban population change is a function of migration, natural increase, and reclassification. In the absence of good migration data (which by necessity includes urban/rural status of migrants), the approach outlined here allows researchers to estimate the contribution of reclassification to the measurement of urbanization processes in a unique way. Additionally, assuming that urban/rural fertility and mortality are known, we can extrapolate estimates of the migration contribution through a residual analysis. For example, if urban population change, urban natural increase, and reclassification are known, the contribution of migration would be the total urban population change less the natural increase and reclassification. While this was beyond the scope of this work, future work might take this step. Second, this research sheds light on the spatial components of a process, which is intrinsically spatial but typically characterized aspatially, that is, by a single summary metric. Through our analysis, we are able to comment on *where, how, to what degree*, and, in some part, *why* reclassification is affecting the measurement of the process of urbanization at a more relevant scale, closer to the operational scale of the underlying demographic processes. Thus, this research provides new directions to more effectively tackle emerging questions regarding coupled nature–human systems and their interactions. Understanding the components of reclassification, particularly in the context of understanding vulnerability (both socioeconomically and environmentally) may have policy implications that can be internalized by agencies and decision-makers [32]. The remainder of this section will discuss our key findings related to these broad statements in finer detail, focusing in particular on matters of time, space, and the built-upness of locations experiencing transition.

It is worth noting up front that, for both census periods, the vast majority of people and census units do not change status over time. Roughly 10% and 7% of the population transitioned from one class to another between 1990–2000 and 2000–2010, respectively. That said, results on the national, census-regional, and census-divisional levels suggest that reclassification is a substantial factor in the measurement of the urbanization process (e.g., roughly 44% and 34% of total urban population growth over each period; see Table 1). Encouragingly, the latter estimate, 34% for the period between 2000–2010, corresponds closely to the estimate of Jiang and O'Neill [9], who found that 27.7% of urban growth could be attributed to reclassification when they decomposed the impact of migration and natural increase (both known during this time period). The magnitude of this effect is influenced significantly by assumptions regarding the timing of reclassification. At the national-level, the difference between assuming reclassification at the beginning versus the end of the time period is just over 40% between 1990–2000 and almost 55% over 2000–2010. As noted earlier, in reality, each block/block-bit that is redefined will technically meet the definition necessary for reclassification at some point during the census period. Here, the most logical hypothesis is to assume that reclassification is distributed across the full period in a relatively uniform pattern. Future work might investigate this hypothesis by looking at the relative population, population density, and connectivity of each block/block bit at the beginning and end of each period, projecting the transition timing of each to generate a more complete picture of the process (over both space and time). (This proposed analysis would be a substantial undertaking, and is well beyond the scope of this work).

Substantial regional variation exists in both the magnitude and consistency of the effect of reclassification on urbanization metrics (see Table 2). For example, in the West Region, reclassification demonstrates a consistently lower impact on the urbanization rate. This trend is likely a function of the relatively high rate of in-migration (e.g., for Colorado or Arizona), indicating that urban growth is occurring as more population moves into the region. Conversely, the Midwest Region and, in particular, the East North Central Division, exhibit a consistently high effect from reclassification. The cities of this division (e.g., Cleveland, Pittsburgh, Detroit) are generally older, exhibiting characteristics of

the "rust-belt", and for some time have been net senders of migrants rather than receivers. As such, the relative effect of reclassification is amplified. In the South Region, the highly populated South Atlantic Division (which includes Florida, North Carolina, and Virginia) most closely resembles the country in general, while the East South Central Division (Alabama, Mississippi, Tennessee, and Kentucky) more closely resembles the rust-belt states, which is not surprising given that these states comprise the bulk of the Deep South and parts of Appalachia, where out-migration is prevalent. Finally, the Northeast Region, and both its component divisions, exhibit inconsistent effects from reclassification over the two census periods (high in the first, lower in the second). Across all regions/divisions it is apparent that assumptions regarding the timing of reclassification remain important and critical for objective uncertainty assessments. Acknowledging the local variation in population change and development, which are spatial in nature, are themes that demographers and other social scientists are accepting as necessary for understanding the consequences of population growth and decline at relevant scales [33].

A final important point, related to one of the key purposes of this work, involves the potential of GHSL to serve as a leading indicator for reclassification. Recall that, in most countries, demographic data at the resolution of census blocks are not available. Moreover, in cases where high-resolution census data do exist, they often lack in temporal resolution (e.g., the number of available censuses) and consistency. The GHSL data offer potential as a partial proxy for underlying demographic phenomena, particularly when they can be used in conjunction with some demographic data, while also contributing important information regarding the built environment, in models of urban growth and change. As indicated by Table 3 and Figure 6, there is clear variation in the built-up characteristics of rural blocks that transition to urban and those that do not. At the national level, rural blocks that transition to urban are some fifty to sixty times more built-up, on average, than those blocks that remain rural. At the divisional-level, these figures vary from 15 to 500 times more built-up (Table S7), and at the city-scale we find blocks in Atlanta that transition from rural-to-urban roughly seven times more built-up than those that do not (in both time periods), while transitioning blocks in Chicago are some 8–10 times more built-up then those remaining rural. The magnitude of the difference likely declines in Metropolitan Statistical Areas (MSAs; cities) because rural land in close proximity to urban areas is likely more built-up, on average, than more remote land. The order of difference identified in the historic data suggest that GHSL could be very useful in identify areas likely to experience reclassification, and including such information in models of urban change may improve our ability to identify such places *and* better quantify the probability of transitioning. While beyond the scope of this work, characterizing the distribution of built-up characteristics across blocks that transition and those that do not would aid in this task. In addition to using GHSL, which is available globally, we could use the Historical Settlement Data Compilation for the Conterminous United States HISDAC-US [34], a new much-longer time-series, for the US, to further determine the timing of reclassification and revise the long-term descriptions of urban growth, nationally, over even longer time periods. Similarly, following blocks across multiple census periods would help to identify patterns of change over time and, potentially, further improve a predictive model.

Several important caveats and limitations require discussion here as well. Changes in block boundaries, particularly when considered within the context of our assumptions regarding the uniform distribution of population within blocks. For example, when boundary changes occur, it is often sometimes the case that blocks are being purposely broken up (or some combination of broken up and amalgamated with others) to separate components displaying urban and rural characteristics. In such a scenario, it may be that the underlying population distribution is a significant factor in the reorganization of boundaries, and thus it is unlikely that the distribution was uniform. In fact, population is likely to have been skewed towards places classified as urban and away from those defined as rural. The result, when applying our analysis to previously rural blocks, would be to overestimate population growth in areas reclassified as urban; whereas, when applied to previously urban blocks, the result might be an overestimate population decline in areas redefined as rural. Given

the relatively small size of blocks in and around urban areas, we believe the impact of this problem on population change estimates would be minimal. However, it is also very likely that urban-to-rural transitions are driven primarily by such reorganizations, and therefore should be assessed with caution. Future work could construct spatio-temporally harmonized enumerated units [35–37] using areal interpolation techniques to overcome the problem of inconsistent block boundaries as a considerable analytical undertaking.

A second related point involves changes in the definition or urban over time. Not only are boundary changes occurring as a function of the underlying characteristics of sub-block areas, but the definitions upon which such decisions are based are in flux in part because the characterization and the underlying meaning of what it is to be urban is also changing. The spatio-temporal nature of this problem is thus further confounded by changing notions as to what we considered urban/rural, complicating the analysis of underlying patterns of change. In this work, we do not attempt to hold the urban/rural definition constant (thus removing these effects) as doing so would represent an extraordinarily difficult exercise (applying the census methodology across tens of millions of block bits as opposed to blocks, which has the potential to change the spatial definition of urban agglomerations, among other effects). Instead, we respect the definition applied at each census period. Subsequently, an unidentified portion of the observed reclassification is a function of definitional change over time.

Finally, as mentioned above, we do not assess the full distribution of population, population change, or GHSL values across each classification. Instead, our analysis focuses primarily on mean values, and, in the case of cities, our analysis focuses primarily of visual assessment. Similarly, although we look at two separate census periods, the analysis is essentially cross-sectional. In the future, we recommend a longitudinal approach that tracks parcels of land over census periods, and an assessment of the full distribution of characteristics for each class of land. Both may contribute to a more robust understanding of urban morphology and thus to improved models of urban change.

## 5. Conclusions

In this paper, we have adopted a new approach—one that is explicitly spatial—that allowed us to address an important and unexplored topic in urban demography: how to estimate the effect of reclassification on the measurement and/or interpretation of urban population change. In particular, we drew attention to the spatial and temporal nature of this change, arguing that both are important. The spatial problem has long been recognized as co-mingling with other factors such as migration, but this paper is the first ever to also call attention to and examine the importance of timing. Additionally, by coupling the spatial approach with a remotely sensed measure of built-up land, we are able to isolate geographic conditions that appear somewhat predictive of future rural-to-urban reclassification. This finding is important for two reasons: first, it offers a potential improvement to spatial models of urban evolution (both population and land), and second because, in countries lacking in highly resolved census-type demographic data, the remotely-sensed data have potential as a proxy indicating the degree to which a given location is urban in nature (again considering both people and land). In regard to this point, it should be noted that this work is specific to the United States, and the process of urbanization is likely to vary substantially across countries with varying demographic, socio-economic, and geographic profiles. Confirming the applicability of this approach will require additional testing across a more complete typology of countries for which good census data are available.

For more than 50 years, demographers have recognized that reclassification may interfere with an understanding of the demographic processes contributing to urbanization [6,9], particularly on the contribution of migration to urban change. With the bulk of the future population growth to take place in the world's cities and towns (particularly in the global south), it is imperative that we find new methods, such as those pioneered here, to decompose the effect of reclassification from the components, timing, and location of urban and city-specific demographic change. Spatial data and tools are now common in demography and the social sciences, and fine-scale spatial data are becoming increasingly

available from national statistical offices around the world, making it inexcusable to continue to give this topic short shrift.

**Supplementary Materials:** The following are available online at http://www.mdpi.com/2071-1050/12/4/1649/s1, Table S1: Aggregate urban/rural population totals and change by transition class, US Census Regions 1990–2000, Table S2: Aggregate urban/rural population totals and change by transition class, US Census Regions 2000–2010, Table S3: Aggregate urban/rural population totals and change by transition class, US Census Divisions 1990–2000, Table S4: Aggregate urban/rural population totals and change by transition class, US Census Divisions 2000–2010, Table S5: Proportion built-up area as detected by the GHSL by transition class, US Census Regions 1990-2000, Table S6: Proportion built-up area as detected by the GHSL by transition class, US Census Regions 2000–2010, Table S7: Proportion built-up area as detected by the GHSL by transition class, US Census Divisions 1990–2000, Table S8: Proportion built-up area as detected by the GHSL by transition class, US Census Divisions 2000–2010.

**Author Contributions:** Conceptualization, B.J., D.B. and S.L.; Data curation, B.J. and D.B.; Formal analysis, B.J., D.B. and S.L.; Funding acquisition, B.J., D.B. and S.L.; Methodology, B.J., D.B. and S.L.; Project administration, D.B.; Supervision, B.J.; Validation, S.L.; Visualization, B.J., D.B. and S.L.; Writing – original draft, B.J., D.B. and S.L.; Writing – review & editing, B.J., D.B. and S.L. All authors have read and agreed to the published version of the manuscript.

**Funding:** The work was funded, in large part, by the US National Science Foundation award #1416860 to the City University of New York, the Population Council, the National Center for Atmospheric Research (NCAR) and the University of Colorado at Boulder, and with additional support from an Andrew Carnegie Fellowship (#G-F-16-53680) from the Carnegie Corporation of New York to D.B.. S.L. also received funding under grant # P2CHD066613 from the Eunice Kennedy Shriver National Institute of Child Health and Human Development to the University of Colorado Population Center (CUPC) at the Institute of Behavioral Science as well as support through a CUPC seed grant The funders had no role in study design, data collection and analysis, decision to publish, or preparation of the manuscript.

**Acknowledgments:** We thank Brian O'Neill, Mark Montgomery, Anastasia Clark, and Leiwen Jiang for comments on an early draft.

**Conflicts of Interest:** The authors declare no conflict of interest.

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
