# Peer review of "Urban Change in the United States, 1990–2010: A Spatial Assessment of Administrative Reclassification"

_sustainability, doi:10.3390/su12041649_

Round 1

Reviewer 1 Report

I do not have any other comments on my side. 

The paper can be accepted in the present form.

Author Response

We thank the reviewer for taking the time to assess our work.

Reviewer 2 Report

This manuscript presents an analysis that quantifies the effect of urban area reclassification on population estimates at multiple spatial scales. The manuscript is clearly structured, the analysis is explained and illustrated well, and the conclusions are supported by the evidence presented. I think the paper makes a nice contribution and should be accepted after making a few minor changes and corrections, which I detail below.

It could made be more clear in the abstract and introduction that you are conceptualising urbanisation primarily in terms of increased population density rather than changes in urban morphology (although there is some discussion of this, it is not really the focus). Making that more clear up front would help readers to decide whether the paper focused on their interest without reading a few pages in. Figure 1: I don’t see rural settlements in the diagram, but it appears in the legend? Also you are discussing in situ migration (do you instead mean in situ urbanisation), but it isn’t labeled in the digram. I think this is referring to the polygon labeled ‘surpass urban population threshold’, but it might be good to use terminology consistently. I am curious why you did not also include the NYC example in the land cover analysis? The legends in Figure 3 are really too small to be legible in the pdf, especially the font size of the symbol labels. Even with my reasonably good eyes, I found them difficult to read. Figure 6: red/green may be problematic for colour vision impaired readers. Why not match the colours you used in Figure 3 for those same categories? It seems like this paper should perhaps be discussed somewhere in the manuscript: Chen, Q., & Song, Z. (2014). Accounting for China's urbanization. China Economic Review, 30, 485-494.

Minor corrections

Abstract: …why reclassification is affecting the process of urbanization at… p. 1 footnote: The term administrative classification… p. 3 …better understanding of the urbanization process in general. p. 3 use country-level data. I think here you actually mean county-level data? p. 4 …delineated by both human-made and physical… scattered places throughout the text: you often refer to a ‘portion’ of the block. Would it be better to describe this as a ‘proportion’? p. 6: in your discussion of Figure 2, it would be better to capitalise the three letters (A, B, C) - the lower case a can be confused with the word ‘a’, and I think the capital letters are more easily readable in the running text. p. 7 …effects of reclassification on measuring … p. 8: weigh heavily: on what? the outcome of the estimation process? p. 9: (1) urban at the beginning of the time period, and (2) transitioned to urban during the time period. I think you mean they were rural at the beginning?!? p. 9: …the assumption made about timing can be… p. 11: …In the later period, which does include the financial crisis that hit the region… ubiquitous (in a few places in the manuscript): I think you mean homogeneous. p. 13: …(substituting 2015 for 2010)… —> I think you mean 2014 not 2015? p. 13: reference to Table 2. But Table 2 isn’t showing rates of population growth. It’s showing relative contribution of reclassification to growth. Did you really mean to refer to that table here? If so, that needs more explanation. p. 15: …growth/change are particularly… p. 16: You refer to Table 1, but the figures you are citing are in Figure 2. p. 16: …The order of magnitude difference identified in the historical data suggest that GHSL could be very useful in identifying... p. 17: …often sometimes the case.. —> it’s either often or sometimes, not both! p. 17: …are being purposefully… p. 17: … might be an overestimate of population decline… p. 18: …In regard to this point…
